# Physical Activity and Exercise Experience in Spanish and US Men with Fibromyalgia: A Qualitative Cross-Cultural Study

**DOI:** 10.3390/ijerph20186731

**Published:** 2023-09-07

**Authors:** Pilar Montesó-Curto, Loren Toussaint, Angela Kueny, Ilga Ruschak, Shannon Lunn, Lluís Rosselló, Carme Campoy, Stephanie Clark, Connie Luedtke, Alessandra Queiroga Gonçalves, Carina Aguilar Martín, Ann Vincent, Arya B. Mohabbat

**Affiliations:** 1Primary Care, Catalan Healthcare System, 43500 Tortosa, Spain; 2Department of Medicine, Rovira I Virgili University, 43201 Reus, Spain; 3Department of Psychology, Luther College, Decorah, IA 52101, USA; touslo01@luther.edu; 4Department of Nursing, Luther College, Decorah, IA 52101, USA; angela.kueny@luther.edu; 5Internal Medicine Unit, Sant Pau i Santa Tecla Hospital, 43880 Tarragona, Spain; ilga.ruschak@urv.cat; 6Faculty and Department of Nursing, Rovira I Virgili University, 43002 Tarragona, Spain; 7Research Division, United Hospital Allina Health, St. Paul, MN 55102, USA; lunnsh01@gmail.com; 8Santa Maria Hospital, 25198 Lleida, Spain; lrosello@comll.cat; 9Faculty of Nursing and Physiotherapy, Lleida University, 25198 Lleida, Spain; carme.campoy@infermeria.udl.cat; 10Mayo Clinic, Rochester, MN 55901, USA; drclark@bsmpartners.net (S.C.); luedtke.connie@hotmail.com (C.L.); vincent.ann@mayo.edu (A.V.); mohabbat.arya@mayo.edu (A.B.M.); 11Research Support Unit, Terres de l’Ebre, Jordi Gol Primary Healthcare University Institute, 43500 Tortosa, Spain; aqueiroga.ebre.ics@gencat.cat (A.Q.G.); caguilar.ebre.ics@gencat.cat (C.A.M.); 12Family and Community Medicine Educational Unit, Tortosa-Terres de L’Ebre, Catalan Healthcare System, 43500 Tortosa, Spain; 13Evaluation Unit, Terres de l’Ebre Primary Care Division, Catalan Healthcare System, 43500 Tortosa, Spain

**Keywords:** exercise, fibromyalgia, men, physical activity, experience

## Abstract

Physical exercise is an indispensable element in the multidisciplinary treatment of fibromyalgia syndrome (FMS). The present study examined if men diagnosed with FMS engaged in any type of physical activity or exercise, the perceived effects from exercise, and who specifically recommended exercise. A qualitative cross-cultural study was performed in fibromyalgia clinical units in Spain and the United States. A total of 17 participants, 10 men from Spain and 7 men from the US, were included. In Spain, a focus group was completed in two parts, one month apart in 2018. In the US, five individual interviews and one joint interview with two men were completed in 2018. Three central themes appeared in the qualitative data: (1) Understanding what constitutes physical activity or exercise, (2) Facilitating or discouraging the performance of physical exercise, and (3) Effects of physical activity or exercise on psychological and social symptoms. The actual practice of exercise by patients with FMS is often perceived as leading to pain and fatigue, rather than a treatment facilitator. Physical activity and exercise can provide benefits, including relaxation, socialization, and increased muscle tone. However, minor opioids limit physical activity as they cause addiction, drowsiness, and decrease physical activity in Spanish men. Recommendations in a clinical setting should incorporate exercise as well as physical activity via daily life activities.

## 1. Introduction

There is presently no singularly effective treatment for fibromyalgia syndrome (FMS) [1]. Rather, the biopsychosocial etiopathogenesis requires a multidisciplinary treatment approach including pharmacotherapy, psychotherapy, and exercise [2,3]. Patient-tailored exercise is important [4,5]. Exercise elicits centrally mediated pain inhibition [6] and decreases muscle stiffness [7], both of which are key benefits in FMS.

There is little FMS research in men [8]. In the past, a qualitative study of men, led by Paulson et al. [9], stated that men experience delays in receiving primary and follow-up care. Exercise research in men is often limited by small, biased, and unblinded trials, among other limitations [10]. Nonetheless, compared to healthy controls, we know that men with FMS are less physically fit [11], have a slower gait speed, and shorter stride length and cadence [12]. Men with FMS have lower perceptions of health status and more physical limitations [13].

Physical activity is a bodily movement produced by skeletal muscles that results in energy expenditure. The term “physical activity” does not require or imply any specific aspect or quality of movement. The term encompasses all types, intensities, and domains [14]. Exercise is a physical activity that is planned, structured, repetitive, and designed to improve or maintain physical fitness, physical performance, or health. Exercise, like physical activity, encompasses all intensities. Physical activity is a broader category, and physical exercise is one specific type of physical activity [14,15].

The American College of Rheumatology (ACR) [16] established guidelines for the use of aerobic exercises in the treatment of adult patients with FMS over the age of 18 years. Brosseau et al. [17] support the use of regular activity for the general management of FMS. Aerobic and muscle-strengthening exercises are effective ways to reduce pain and improve overall well-being in people with FMS [18,19]. Aquatic training and Tai Chi are also examples of exercises that benefit FMS patients [10,20].

The Rheumatology’s Spanish Society (SER) formal recommendations state that there are insufficient studies to recommend the use of nonsteroidal anti-inflammatory drugs (NSAIDs) for the treatment of pain in adult patients with FMS [21]. The use of major opioids to treat pain in patients with FMS is not recommended because of insufficient evidence for efficacy and the possible risk of adverse effects. Furthermore, data on pharmacological treatment in patients with FMS have shown poor results, with only a 33% improvement in the severity of the variables analyzed in one third of patients [21]. A reduction in symptom severity of greater than 50% over baseline was considered significant in the SER guidelines [22].

The benefit of exercise for men with FMS is an understudied topic. In one study, six men with FMS followed a Tai Chi program for four months. The intervention improved physical flexibility, mental health, pain, aerobic capacity, agility, dynamic balance, and overall FMS symptoms and impact [23]. In another study, two men with FMS continued to maintain their normal physical and psychosocial functioning four years after a physical exercise program [24].

The purpose of the present study was to explore if men diagnosed with FMS in the United States (US) and Spain engaged in any type of physical activity or exercise, and if so, its frequency, intensity, and perceived effects from exercise. Furthermore, this study sought to understand who specifically recommended the exercise as a means of treatment and whether exercise was performed under supervision or done privately.

## 2. Materials and Methods

### 2.1. Design

This study is a qualitative cross-cultural study from an interview guide in Spain and the US. Men with FMS provided qualitative responses on their physical activity and exercise experiences, offering an opportunity for a comprehensive analysis. This study is part of a larger study on FMS in men, where different aspects of this disease were studied by the Rovira i Virgili University Foundation. In this article, we focus specifically on the perception of physical activity. An identical interview guide was used in patients with FMS in Spain and in the US, which consisted of four main questions. The first question from the first focus group explored perceptions, signs, symptoms, and the impact of FMS. The second question explored feelings, reactions, and biopsychosocial resources for coping with FMS. The third question explored the personal impact or the consideration of any gender experiences they might have identified. The fourth question explored physical exercise performance and its effects on health. Within the study, participants also completed surveys focused on demographics, physical activity, and the impact of FMS symptoms, in order to provide comparative analysis across samples in the US and Spain. This article primarily focuses on the qualitative results but also reports on some of the survey results to provide context to their qualitative experiences. The research team was composed of healthcare professionals and professors, of whom 3 are men and 10 are women.

### 2.2. Participants and Setting

Sampling was purposeful, which is a widely used technique in qualitative research for the identification and selection of information-rich cases for the most effective use with limited resources. Information-rich cases are those from which much can be learned about issues of fundamental importance to the research objective [25]. Maximum variation sampling was used to purposively select a heterogeneous sample of men with FMS and therefore ensured a small sample of broad diversity [26]. In our study, we selected participants from different age groups, areas, and countries to obtain a broader perspective.

### 2.3. Data Collection

A semistructured interview guide (Figure 1) was developed that included predetermined topics with open-ended questions, as well as complementary questions to allow for clarification and exploration of responses in greater detail [27].

We obtained a total of 17 participants, 10 from Spain and 7 from the US. The participants were chosen from individuals receiving FMS care at different clinical units in Spain and the US. All interviews and focus groups were held at the participating clinical sites (US and Spain). Men were included in this study if they were older than 18 years of age and had a diagnosis of FMS. Of note, gender identity is in continuous transformation and categorization; for the purpose of this study, we will only refer to the biological sex at birth [28]. To be included in the study, participants had to have fulfilled the 2010 American College of Rheumatology criteria [16] for FMS, signed the informed consent, and have been willing to participate in the study. The exclusion criteria consisted of men who were outside the age range, had a history of neurological pathology or traumatic brain injuries, or were diagnosed with dementia or schizophrenia.

#### 2.3.1. Spain

Participants were selected from a list provided by the Fibromyalgia clinic from Santa Maria University Hospital (Lleida, Catalonia, Spain). The Central Sensitivity Syndromes (SCS) Unit–Fibromyalgia, Chronic Fatigue and Multiple Chemical Sensitivity Syndrome-of the University Hospital Santa Maria de Lleida has been operational since 2008. It assists an average of 4000 patients per year among all the professionals: 2 rheumatologists, 1 nurse, 1 physiotherapist, and a psychologist. In addition to outpatient consultations, there are therapeutic groups and training in other specialties. There is the possibility of hospitalization in serious cases as it is a 400-bed hospital, and the clinical unit conducts its own research within the framework of the Biomedical Research Institute of Lleida and also with other research groups. Two focus group sessions were conducted in Spain, each covering two of the above questions. This allowed participants to elaborate on each topic in greater detail. A focus group consisting of 10 patients diagnosed with FMS was organized. Groups were completed in two parts, one month apart in 2018 (May and June). Each group interview lasted around 120 min, was audio-recorded, and was transcribed afterward by two of the researchers who witnessed these meetings. They were led by two health professionals trained in focus groups and a third observer member of the research team.

#### 2.3.2. United States

Participants were chosen from a list provided by the Fibromyalgia and Chronic Fatigue Clinic at the Mayo Clinic (Rochester, MN, USA) and from voluntary contact in response to public regional advertisements of the study in Decorah (Iowa, IA, USA) in August 2018. The Fibromyalgia and Chronic Fatigue Clinic at the Mayo Clinic has been operational since 1999. On average, the clinic provides approximately 1000 patient consultations annually. All patients seen in this clinic are seen on an outpatient referral basis, which is consistent with patients from across the country and globe. The clinic provides a consultation with an FMS specialist, as well as an 8-h educational program, focusing on pathophysiology, medication management, and nonmedication treatment strategies. Due to the logistical difficulties of bringing a group of men together, individual interviews were the best option for this cohort. Two men completed a joint interview while five men completed individual interviews. Two health professionals led the interviews, both trained in interview techniques and familiar with the study. Individual interviews lasted 45–60 min. Interviews were recorded for accuracy and transcribed afterward.

Prior to the first focus group or interview, we provided the participants with an ad hoc questionnaire to collect the participants’ sociodemographic characteristics and clinical data, including questions about physical activity or exercise.

As a panel of experts, we assessed the face and content validity of the questions before initiating data collection. Saturation data were achieved with these focus groups. Member checking was completed for each interview; the researchers reflected to participants their responses to ensure an accurate representation of their intent. Four semistructured questions also allowed men to report who referred them to exercise as a treatment and whether the exercise was supervised (exercise as a type of prescription for which they need to plan the dose and monitor the amount and track activities):If you do physical activity or exercise, explain what type(s) you do and if you supervise the exercise you perform.If anyone has recommended to you to perform physical exercise, what person or professional has that been?If the exercise you perform is supervised by any person and/or professional (gym, coach…).If physical activity or exercise reduces (or not) pain, fatigue, or any other symptoms. Describe the benefits or harmful effects in detail.

### 2.4. Data Analysis

Qualitative content analysis was facilitated using Atlas.ti Version 8, and all codes were translated into English. Inductive content analysis was used to examine the transcripts for recurring patterns or themes [29]. The standard practice is to allow categories to emerge from the data; that is, by researchers who immerse themselves in the data and allow new ideas to emerge [30]. Biopsychosocial and gender perspectives informed the analysis [31,32].

### 2.5. Rigor

The rigor of the analysis was ensured by applying the Lincoln and Guba criteria for qualitative reliability [33].

Credibility: The first and third researchers were involved in data collection, analysis, and reporting of the findings; both were experts in qualitative methodology. Each of these researchers was assisted by a Ph.D. student in Spain and a nursing student in the US. All authors contributed to the writing of the manuscript and read, reviewed, and approved the final draft of the manuscript.Transferability: The transferability of the study is derived from a detailed description of the context of the study which were specialized units of fibromyalgia and confirmed diagnosis in all the volunteers. All those who agreed to participate did not drop out. The unification of the categorization and analysis process by the research team brought to light the experiences of these men with FMS that may resonate with other men with FMS.Reliability: The codification and categorization were carried out by the four researchers, working in pairs. Each pair of researchers in the US and Spain coded each transcript together to ensure consistent coding. Together, the four researchers built code definitions and reviewed coding decisions with each other in frequent meetings throughout the analysis. In addition, the different and multiple meetings via Skype by the three first members finished defining the final themes, categories, and subcategories. These were approved by the consensus of the whole group. The terms “physical exercise” and “physical activity” were analyzed in depth by 7 researchers, one of them a physiotherapist and Ph.D. in Anthropology, the rheumatologist, the second author professor, Ph.D. in Psychology, and the two authors in senior position, specialists in internal medicine and Ph.D. in Medicine.Confirmability: The original data, including draft and data analysis records, were safely stored for future reference. In addition, to observe the similarity and differences between the two countries, the US researchers translated the text into English and the Spanish into English?

### 2.6. Ethical Considerations

The current study was approved by the Ethics and Research Committee of the Jordi Gol Primary Care University Institute Foundation from Barcelona, code: 4R18/223, Mayo Clinic, IRB 10429.010, and the Luther College Human Subjects Review Board. Informed consent was obtained from all participants.

## 3. Results

### 3.1. Sample Characteristics

In our study, the men ranged in age from 30 to 63 years old, with an average age of 52 years. Most of the men were married, had completed secondary education, and lived with multiple people in their homes. The US men were more actively working/employed than the Spanish men. Table 1 summarizes the patient characteristics and the different types of exercise reported by men in this study across Spain and the US.

Many participants did not experience any reduction in pain or fatigue due to exercise either in the US or Spain. Difficulties in exercising were similar in the US and Spain. The main form of exercise was walking in both cohorts. The recommendation for exercise in Spain came mainly from rheumatologists, while in the US it was from internal medicine physicians or general practitioners.

Table 2 includes the name, country, age, and consumption of opiates. Fictitious names were used to protect participant identity.

### 3.2. The Exercise Experience of Men

Qualitative analysis revealed three main themes: (1) Men’s understanding of “physical exercise”, (2) Facilitating or discouraging the performance of physical exercise, and (3) Effects of physical activity or exercise in symptom, psychological, and social aspects. Table 3 illustrates the categories and subcategories.

#### 3.2.1. Theme 1: Understanding What Constitutes Physical Activity or Exercise

At first, when we asked the question if men performed exercise or not, most men answered no. As the interview progressed, we came to understand that men had different conceptions of the term “exercise”. The following categories were detected: associate the meaning of the term physical activity as physical exercise and associate exercise with physical activity (Table 3).

##### Associate the Meaning of the Term Physical Activity as Physical Exercise (a Sport or Planned Activity with the Objective of Improved Health)

Participants believed that exercising necessarily implies practicing some type of organized activity or sport. So, with that understanding, some of them immediately indicated that they were not physically active. Some men associated physical activity with physical exercise. However, fewer Spanish men continue exercising after being diagnosed with FMS. The subcategories identified were: cycling, stretches, treadmill, physical therapy work, swimming, and walking.

##### Associate the Exercise with Physical Activity

Most participants described physical activity as daily life or work activities like taking care of farm acreage, walking, changing a light bulb, and climbing stairs. More US men incorporated physical activity into their daily lives. Walking was the most frequent physical activity and was the most frequently identified form of physical activity/exercise even when participants denied engaging in any physical activity. The subcategories identified were: taking care of the acre, climbing stairs, and walking.

#### 3.2.2. Theme 2: Facilitating or Discouraging the Performance of Physical Exercise

We realized that there were factors that helped or hindered the performance of physical exercise. The categories that emerged were: recommendation of physical exercise by a specialist and opiates affect exercise performance (Table 3).

##### Recommendation of Physical Exercise by a Specialist

Recommendations were made by specialists for men to continue exercise with FMS. The most frequent recommendations were from general practitioners and rheumatologists. US men received more recommendations for physical exercise from healthcare providers. Patients in Spain did not receive recommendations from internal medicine physicians, because they were not visited by them. The subcategories identified were: practitioner, rheumatologist, and internal medicine.

##### Opiates Affect Exercise Performance

Opioids caused drowsiness, limitation of movement and speech, and addiction. These elements decreased exercise performance. Most Spanish men are taking minor opioids, while no US men were on opioids. The opioids restricted physical activity and some of them did not leave the house because of the drowsiness caused by the opioids. Other side effects were nausea and difficulties with speech. One participant was able to wean himself off the opioids by progressively decreasing the dose, because he did not want to go to the hospital, where he would be given methadone. The categories that emerged were: sleepiness, limitation of the movement, skills and speech, and addiction (Table 3).

#### 3.2.3. Theme 3: Effects of Physical Activity or Exercise in Symptom, Psychological, and Social Aspects

Men discussed the difficulty of physical activity because of the symptomatology of FMS, especially fatigue and pain. More harmful than beneficial effects of physical exercise were detected. Pain and fatigue were the main harmful elements observed. Mentally, they felt better. Exercise helped them to channel anxiety and interact with other people. The categories identified were: harmful effects and beneficial effects (Table 3).

##### Harmful Effects

In general, even minimal activity resulted in a worsening of FMS symptoms. Pain and fatigue appear after starting physical activity; for some men, after minimal exercise.

A participant walking only 100 m had to lie down all day to recover. One participant after 20 to 45 min of exercise in the garden had worse cramps. Another participant, after 1 h of physical activity mowing lawns, felt bad for two weeks. Another patient stated that stretching was painful, then he felt better, but the pain returned. The subcategories detected were: they get so tired doing activities of daily living, they get tired doing small tasks, their overall strength is worse, fatigue and pain.

##### Beneficial Effects

Many participants noted that they felt worse after exercising because their FMS caused them more pain physically. However, many men noted benefits in cognitive, mental, and social aspects, rather than physical. They do not recognize that exercise helped them physically because their body was affected by the underlying condition and minimal effort made it worse. Nevertheless, for some participants, exercising actually entertained them, and it was used as a sort of distraction, which involved mental relief, escape, and socialization. One participant recognized the benefit of stretches. To decrease anxiety, many participants with FMS performed relaxation exercises that can be combined with medication therapies. Yet also, physical exercise appears as part of holistic health along with good nutrition, sleep improvement, and stress reduction.

Spanish men enjoy socializing when they are walking and meeting people outside; this helps them to have a social connection.

It is important that the person does not raise the level of activity to levels above which recovery will be difficult. Men moderated the exercise and adapted it to their own needs. The subcategories that emerged were: physically, they feel worse, but mentally better, channel anxiety, socialization, moderation of physical exercise is perceived as beneficial, a distraction (Table 3).

## 4. Discussion

Men explained that the lack of physical activity or exercise was not because of a lack of motivation, but mainly due to the symptomatology of the disease, where pain and fatigue were the main barriers. In alignment with the findings in our study, Paulson et al. (2002) [9] and Hooten et al. (2017) [13] pointed out that daily simple tasks such as walking, cooking, making a bed, and shaving, among others, result in increased pain. Despite all the challenges, some men did attribute mental benefits to exercise, but not on a physical level. Furthermore, they reported that exercise such as walking offered a form of entertainment, socialization, and distraction. Distraction relieved their symptomatology, which was consistent with previous findings [34,35]. As a result of physical activity, participants could repurpose their anxiety and could socialize more [36,37].

Most participants from Spain were taking opioids and it caused drowsiness, limitation of movement, and addiction. Men with FMS were prescribed minor opioids in Spain, but not in the US. The Spanish Society of Rheumatology (SER) does not recommend the use of major opioids, yet does not mention the detrimental effects of minor opioids [13]. Moreover, the medical center where the research was carried out follows the recommendations of the 2002 WHO analgesic ladder for chronic pain, specifically the Guide of the Spanish Society of Primary Care Physicians (Semergen AP), Spanish Society of Family and Community Medicine (semFYC) and Spanish Society of General and Family Physicians (SEMG). This guideline recommends minor/weak opioids in the face of insufficient pain control with antipyretic analgesics and nonsteroidal anti-inflammatory drugs [38].

In recent decades, opioids for the treatment of chronic pain have increased dramatically. Yet, it has also been shown that the discontinuation of opioids can lead to an improvement in the clinical manifestations of FMS [39]. Benzodiazepines have also failed to demonstrate efficacy and in the long term are detrimental to the treatment of FMS patients [40]. Furthermore, Johnson et al. [41] observed that addiction to opioids can lead to cold and disconnected human interactions [41].

A strength of our study is using qualitative methodology to explore the experience of what constitutes exercise and physical activity for men with FMS. Important elements appear to be considered when clinicians provide recommendations or health education to improve their health. Physical exercise is an important part of treatment that should be incorporated into the treatment of these patients. Maintenance of exercise is essential for the improvement of functionality for men with FMS [2,42]. Following this idea, Poindexter (2017) [43] suggests that nonpharmacological treatments should be considered in tandem with medications. These treatments may include aerobic exercise, acupuncture, massages, and the use of psychosocial therapies.

Another strength of the study was the use of a cross-cultural approach, which allowed for the identification of different factors influencing the performance of physical exercise. In our study, fewer men from Spain engaged in physical exercise after a FMS diagnosis. More patients in the US have incorporated exercise into their daily lives, most of them following a training program that emphasized the importance of maintaining an exercise program. Fewer people in Spain incorporated exercise into their daily lives; this is likely due to differences in resources, culture, and healthcare policy. Also, US men received more physical exercise recommendations from healthcare providers. However, men from Spain perceived more benefits of physical exercise in terms of socialization.

We believe that the focus groups and group interviews were of high importance for the identification of these aspects as well as the effects of physical exercise such as tiredness, decreased strength, fatigue, and pain. Some benefits also appeared, such as mental, cognitive, and social improvement. There may have been some differences in the results due to the specific interview methods utilized (focus groups in Spain and interviews in the US). According to Guest et al. [44], sensitive and personal information is often generated in a focus group format, while individual interviews provide richer and more detailed information. In our study, we did not observe a difference in the richness or sensitivity of personal information in the data collection and analysis. The use of the same semistructured questions, and the use of one research personnel with expertise in qualitative research in Spain and another in the US, contributed to greater uniformity in the data collection procedures across sites. Finally, the frequent meetings of the research team before and during the entire project (face-to-face and virtual) helped to homogenize the entire process of project design, data collection, and analysis.

It should be noted that Spain and the US have two very different healthcare models. Spain has the Beveridge model, and the US has the Medicare and Medicaid model. Also, in the US there is no universal healthcare coverage. Normally, large companies cover, at least partially, the cost of health insurance for full-time employees. The Beveridge healthcare model originates from the British health system and is followed in Sweden, Finland, Norway, Denmark, Iceland, Spain, Italy, and Portugal, among others. It is financed entirely by taxes and provides universal health protection via a wide network of health centers, usually with no co-payment for health services, except for pharmaceuticals [45]. Although Spanish patients believe that it should be the public health system that provides them with physical exercise, instead of themselves, once the services of doctors are accessible to the entire population, many patients, especially those seeking rehabilitation programs, experience long waiting times and short-term therapeutic programs.

The understanding of what participants with FMS consider to be physical exercise goes beyond planned physical activities to maintain health. Also included in exercise are activities of daily living that are not initially recognized as physical activities but are of vital importance.

Healthcare professionals are a key element in increasing physical activity by including exercise in their recommendations. They should emphasize not only sports activities or planned activities to maintain health, but also those that incorporate physical activity into activities of daily living such as gardening, climbing stairs, or walking. Conversely, opioid use has been shown to be a limiting factor in physical exercise.

Future studies could explore how recommendations on moderate physical exercise and physical activity, tailored to the individual, could improve satisfaction with physical and mental health and socialization, decrease exercise dropout and opioid prescription, and enhance exercise self-responsibility.

The findings of this study can be applied in the development of health education programs, rehabilitation, primary care, research studies, and updating of protocols and clinical guidelines for the assessment and treatment of FMS in various healthcare settings.

### Limitations

This study has some limitations. First, the sample size was small. In addition to the fact that fewer men are diagnosed with FMS, we hypothesize that many did not want to make themselves visible in their communities, due to the associated stigma associated with FMS. When researchers called men, they told them, in confidence, that they did not want their disease to be made public. Even just the possibility of participating and making their diagnosis known was a significant deterrent to participation.

Second, the Spanish data were collected with the focus group methodology, while the US data were collected using an individual interview method. Hence, there are cross-cultural methodological differences, even though the qualitative measures collected were exactly the same.

## 5. Conclusions

Although physical exercise was prescribed by rheumatologists, physiotherapists, family doctors, and internal medicine specialists, half of the men with FMS did not practice any physical exercise. Rather, men with FMS engaged in physical activity as part of their daily lives and work. Among the reasons why participants did not engage in physical exercise was mainly due to the perceived fatigue and pain. However, physical activity and exercise brought about benefits, including relaxation, socialization, and increased muscle tone.

Recommendations for men with FMS to increase their activity should focus on ways for them to incorporate it into their daily lives. Clinicians should be cautious and emphasize moderation in exercise to avoid the iatrogenic effects of exercise in the treatment of FMS. Furthermore, healthcare providers should be aware of the negative effects of opioids on physical activity and exercise.

## Figures and Tables

**Figure 1 ijerph-20-06731-f001:**
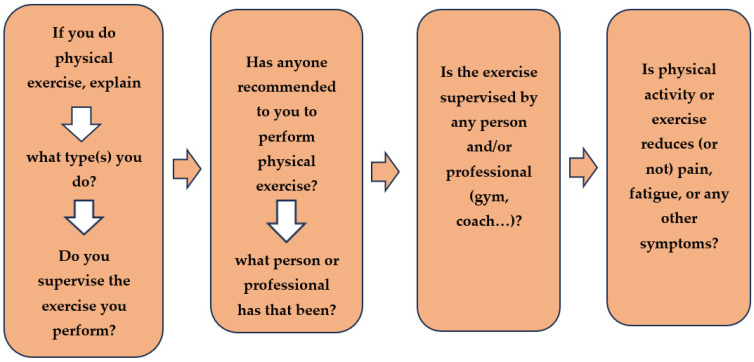
Semistructured interview.

**Table 1 ijerph-20-06731-t001:** Patient sociodemographic and exercise characteristics.

Variable	United States (*n* = 7)	Spain (*n* = 10)
Civil status		
Married	6 (USP1, USP2, USP3, US P5, US P6, P7)	5 (SP2, SP3, SP4, SP5, SP9)
Single	1 (USP4)	1 (SP7)
Divorced/Separated	0	3 (SP1, SP6, SP8)
Widowed	0	1 (SP10)
**Education level**		
No high school	0	1 (SP4)
High school	3 (USP1, USP4, US P5, USP6)	7 (SP1, SP2, SP5, SP6, SP7, SP9, SP10)
Four-year college	2 (USP7)	2 (SP3, SP8)
Graduate/Doctoral degree	2 (USP2, USP3)	0
**Occupational status**		
Active worker	4 (USP1, USP2, USP3, USP7)	0
Unemployed	1 (USP6)	3 (SP1, SP5, SP7)
Active w/work disability	0	2 (SP2, SP4)
Permanent disability	2 (USP4, USP5)	2 (SP8, SP9)
Retired/pensioner	0	3 (SP3, SP6, SP10)
**Number of people living in the home**		
One	1 (USP4)	2 (SP1, SP5)
Two to four	6 (USP1, USP2, USP3, USP5, USP6, USP7)	8 (SP2, SP3, SP4, SP6, SP7, SP8, SP9, SP10)
**Notes the pain and fatigue decrease with exercise**		
Yes	1 (USP5)	0
No	6 (USP1, USP2, USP3, USP4, USP6, USP7)	8 (SP2, SP3, SP4, SP5,S p7, SP8, SP9, SP10)
Depend (temperature water or exercise intensity)	0	2 (SP1, SP6)
**Types of exercise**		
Walking	5 (USP1, USP2, USP3, USP4, USP7)	7 (SP2, SP3, SP5, SP6, SP8, SP9, SP10)
Swimming	0	2 (SP1, SP6)
Yard work	4 (USP1, USP5, USP6, USP7)	0
Farm work	1 (USP1)	0
Stretching	2 (USP3, USP5)	1 (SP7)
Elliptical	1 (USP4)	0
Biking	2 (USP3, USP4)	0
Shoveling	1 (USP1)	0
Digging	1 (USP1)	0
Stairs	1 (USP5)	1 (SP3)
Treadmill	1 (USP3)	0
Manual labor	2 (USP1, USP6)	0
Run	0	0
Pilates	0	1 (SP7)
Aqua gym	0	1 (SP6)
**Who recommended physical exercise?**		
Rheumatologist	1 (USP6)	6 (SP1, SP2, SP3, SP6, SP8, SP9)
General Practitioner	4 (USP1, USP4, USP5, USP6)	3 (SP1, SP6, SP8)
Physiotherapist	2 (USP1, USP4)	4 (SP3, SP4, SP6, SP9)
Traumatologist	0	1 (SP9)
Psychiatrist	0	1 (SP9)
Psychologist	0	1 (SP9)
Internal Medicine	4 (USP1, USP2, USP3, USP7)	0

**Table 2 ijerph-20-06731-t002:** Fictitious name, country, age, and consumption of opiates.

Participant	Name	Country	Age	Opiates
1	Daniel	Spain	59	Yes
2	Jack	Spain	30	Yes
3	Samuel	Spain	46	No
4	Alexander	Spain	45	Yes
5	Adam	Spain	60	Yes
6	James	Spain	55	Yes
7	Jordan	Spain	35	No
8	Jonathan	Spain	55	Yes
9	Julian	Spain	50	No
10	Victor	Spain	53	Yes
1	Stephen	US	57	No
2	Don	US	51	No
3	Andrew	US	57	No
4	Matthew	US	50	No
5	Oliver	US	63	No
6	Ryan	US	60	No
7	Henry	US	53	No

**Table 3 ijerph-20-06731-t003:** Themes, categories, subcategories, and quotes on physical exercise in men with FMS.

Themes	Categories	US Subcategories	Spain Subcategories
**1. Understanding what constitutes physical activity or exercise**	**Associate the meaning of the term physical activity as physical exercise** (a sport or planned activity with the objective of improved health)	**Cycling:** *‘I ride a bicycle for 20 min; I take a couple of laps before and after’. (Andrew-P3, US)* **Stretches:** *‘I do stretches… my own stretches. I learned it here (Mayo) in the program which I am going to implement’. (Oliver-P5, US)* **Treadmill:** *‘I run on a treadmill for 20–25 min. It is a low-key exercise, so it does not intensify or contribute to more pain’. (Andrew-P3, US)* **Physical therapy work:** *‘That was for a separate hip issue and that seemed to help quite a bit. I guess that is a potential resource for the future for sure’. (Don-P2, US)* ***Walking:*** *‘I could walk for 2 h, I think. Only real strenuous activity where you’re sweating, and your heart rate is up that…I firmly believe that I need some exercise and although the Mayo Clinic told me I should walk for 10 min I believe I could do it for 2 h without any problem’. (Henry-P7, US)*	**Cycling:** *‘I liked cycling a lot, but now with vertigo, I had to leave it’. (Julian-P9, S)* **Swimming:** *‘I think swimming is the best but now I can’t do any exercise at all’. (Jack-P2, S)* **Walking**: *‘In fact, I was recommended that before I get tired, I should calculate when I get fatigued, and before that I should stop. The problem is that I must stop every 15 min, but I try to force myself a little to improve and try to get to 20 min.’ (James-P6, S)*
	**Associate the exercise with physical activity** (**daily life activities**)	**Take care of acreage:** *‘I overdo physical exercise, try to do a little more, I push myself as much as I can. I must limit my activity to 30 min. Whatever physical things I do is whatever I do regarding taking care of our acreage. There is yard work, 15 or 20 pounds… I can do a little more than that, but I need to plan to get my legs under… it is a whole effort’. (Ryan-P6, US)* **Climb stairs:** *‘I do not have an exercise program, but at home I have stairs… I am up and down those about 15 or 17 times a day’.* (**Oliver-P5 US**) **Walking:** *‘I like to do walking and because I need to lose weight, but the more I do the more I hurt’. (Stephen-P1, US)*	**Walking:** *‘I don’t practice exercise. I have tried to do it, but after 10 min of walking, I can’t do it anymore. But even if I do, of course, I live in the suburbs, I have 2 girls, I take them up and down, I take them up and down, I take care of them all day long. During the day I do not stop, this says (clock) that today I did 8 km more or less’. (Samuel-P3, S)*
**2. Facilitating or discouraging the performance of physical exercise**	**Recommendation of physical exercise by a specialist**	**Practitioner:** *‘They (providers) have all done it or recommended it because of my weight’. (Steven-P1, US, and Mattew-P4, US)* **Rheumatologist:** *‘The Dr. … did… the rheumatologist up there. I think my rheumatologist did too and my family internist back home… [they] recommended it and there was no implementation on what to do’. (Ryan-P6, US) ‘I think my rheumatologist did too and my family internist back home…’. (Oliver-P5, US)* **Internal Medicine:** *‘What they were suggesting was that maybe you should try to walk for 10 min’. (Henry-P7, US) ’I have been recommended the exercise bike, and elliptical trainer by professionals of Internal Medicine and rheumatology. I have been visiting a chiropractor for many years’. (Don-P2, US)*	**Practitioner:** *‘I went swimming because doctors recommended it to me for the FMS, but I do not know if I am a rare species…’. (Daniel-P1, S, and Mattew-P4, US)* **Rheumatologist:** *‘I want to talk with the rheumatologist to see if he prescribes me a physiotherapist to see if he can fix these legs’. (Victor-P10, S)*
	**Opiates affect exercise performance**		**Sleepiness***: ‘I take the tramadol, it leaves me drowsy, and I can’t do any activity, but the next day I wake up normal and spend the day acceptable’. (James-P6, S)* **Limitation of the movement skills and speech:** *‘I had to give up everything I was doing during all these years. I was a jumper coach, I had been in the high-performance center, I had been runner-up in Spain 2 times, third in Spain in jumps with pole-vaulting, then after 7 years without jumping, I was 3rd in Catalonia. With the medication I take, I can’t do any physical activity. At first, I was taking Palexia (tapentadol), but I was very nauseous, and they changed it to tramadol. They gave me tramadol intravenous in the hospital because the oral medication did nothing for me’. (Jack-P2, S) ‘I’ve gone to pick up the girls at a loss and could barely talk. I’ve been through 3 pain clinics, and I can assure you that there is nothing that takes away the pain. Morphine patches 100 mg. for 2 years and they don’t take away the pain. I eliminated them myself for 8 months by cutting off a tiny bit every day because otherwise I had to go to the hospital to take methadone and it didn’t make any sense’. (Samuel-P3, S)* **Addiction:** *‘And I say yes, I am hooked on opiates. I tried to stop the medication, I told him everything and I gave up everything and I couldn’t, in 3 days I needed it and I had a withdrawal that I couldn’t with. And then, after all, you’re useless, you’re lying in bed because you’re too high to go out, to be able to talk to people… (Jonathan-P8, S)*
**3. Effects of physical activity or exercise in symptom, psychological, and social aspects**	**Harmful effects**	**They get so tired doing activities of daily living:** *‘I try to take 20 min into 45, then there is fatigue and brutal joint pain for the rest of the day and into the next day… More labor that I do the worse my cramps are. It hurts and when it hurts you stop’. (Ryan-P6, US). ‘This weekend I was running a chainsaw and sweating a ton and just trying to work normal…and I was working with a lawn mower for 1 h and then I was sick for two weeks’. (Henry-P7, US)* **Their overall strength is worse:** *‘Not any improvement, usually it’s worse. ‘I force myself but usually it causes pain. During and afterward, but usually afterward’ (Mattew-P4, US). ‘My overall strength is probably down 60–70% from what it was a year ago’ (Oliver-P5, US and Ryan-P6, US)* **Fatigue and pain:** *‘The more I do the more I hurt. Then you do not walk, and you do not do things because you are hurt. It is a big snowball effect. I could not do anything they wanted me to do (yoga). It was very hard, especially for the joints’ (Stephen-P1, US). ‘I do not think it helps with my level of fatigue. I still wake up tired’ (Andrew-P3, US). ‘Stretching is painful but it feels better after and then it comes back. I do that 5–10 times a day’ (Oliver-P5, US). ‘It actually makes the fatigue and pain worst’ (Don-P2, US)*	**They get so tired doing activities of daily living:** ‘I park the car to take the kids to school and I’m already broken, I arrive at the car, and I feel my legs that I cannot take anymore’. (Samuel-P3, S) **They get so tired doing small tasks:** *‘After a shower, I need 15 min to recover from the effort that is involved’ (Jonathan-P8, S). ‘If I walk 100 m to take a coffee with a friend when I return home, I lie down in bed all day’ (Victor-P10, S). ‘You try to change a light bulb, you raise your arm and it hurts all day’ (Adam-P5, S)* **Their overall strength is worse:** *‘I tried to do everything, but after 10 min walking, I cannot take it anymore’ (Samuel-P3, S)* **Fatigue and pain:** *‘If I start peeling 4 potatoes, hair 2 and I have to wait to peel the other 2. ‘If you walk you have pain, but at 50 steps you have fatigue too’ (Samuel-P3, S). ‘There are moments you cannot breathe, there are times when you can walk a lot, others you cannot walk, the classic FMS pain…’ (Adam-P5, S). ‘I have the beard on purpose, so I do not have to shave myself, because it represents a great effort for me’ (James-P6, S). ‘After the minimum exercise I feel worse’ (Victor-P10, S)*
	**Beneficial effects**	**Physically, they feel worse, but mentally better:** *‘And I do get pain doing it, but the pain level doesn’t increase to a point where I can’t deal with it. The benefit is felt fresher afterward, my mind feels cleaner. That is why I continue to do it. It benefits me more cognitively’ (Andrew-P3, US). ‘The more physical labor that I do the worse my cramps are’ (Ryan-P6, US)* **Channel anxiety: ***‘So, there was a fear of that at the time and the anxiety that I am stuck with this pain for the rest of my life. You know, what am I going to do? How am I going to deal with this stuff? …they give you exercise programs to relax your muscles, but they don’t really target fibromyalgia…If you teach them how to deal with the pain and you have some medication that can work, and you combine the two together then you can really help someone deal with it and to live a very positive lifestyle’ (Andrew-P3, US)* **Moderation of physical exercise is perceived as beneficial:** *‘On a good day, you must watch yourself that you do not do too much with that good day. You must cut back on what you do otherwise… you pay for it 2 o 3 days down the road. ‘I realized that I had two options: to do nothing and stay at home complaining or, even if I felt tired and with pain, but living my life. A while ago, in the hospital we made a study about our heart rate and how long we could walk, for me it was a very strong rhythm, I tried to do 3 or 4 days. If I do 20 or 30 min, then I have to be resting triple’ (Ryan-P6, US). ‘You really have to taper it, moderation’ (Oliver-P5, US). ‘I go up and down steps, 15 to 17 times a day, then do my own stretching. I also do some yard work, shopping and driving the car. That’s all, I have no program’. ‘The more physical work I do, the worse my cramps become’ (Oliver-P5, US)*	**Physically, they feel worse, but mentally better: ***‘I’m tired, but I feel better. Physically I’m worse, but mentally I’m better’ (Jonathan-P8, S). ‘One must force oneself to do something, so the mind becomes distracted’ (Samuel-P3, S).* **Channel anxiety: ***‘Maybe swimming is what worked best to channel anxiety, but now I cannot channel it with anything’ (Jack-P2, S)* **Socialization: ***‘The important thing is to get out of the house because I socialize, but if I walk more than 20 min, I find myself three times as tired. I have to stop every 15 min then; I try to force me’. If I leave home and I meet someone, I feel better, but then I’m sitting 2 days at home, and I do not do anything else’ (James-P6, S). ‘I have a 4-year-old dog and I leave the house 1 or 2 times a day when I am well. There are days when I can’t, but I admit that it helps me to get out of the house and to talk to some people I meet’ (Julian-P9, S)* **Moderation of physical exercise is perceived as beneficial:** *‘In theory, you do not have to get tired, so you do not have to rest’ (James-P6, S)* **Distraction: ***‘If you’re busy, you’re less focused on pain and you feel better’ (Alexander-P4, S)*

## Data Availability

The data used to support the findings of this study are available from the corresponding author upon request.

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
