# Peer review of "Physical Activity and Exercise Experience in Spanish and US Men with Fibromyalgia: A Qualitative Cross-Cultural Study"

_ijerph, 2023, doi:10.3390/ijerph20186731_

Round 1

Reviewer 1 Report (Previous Reviewer 2)

Thanks, I have no further comments.

Author Response

REVIEWER 1

Thanks, I have no further comments.

Reviewer 2 Report (Previous Reviewer 3)

The manuscript under review addresses how physical activity or exercise, its frequency, intensity, and effects are perceived by a sample of men diagnosed com FMS, and if there would exist differences on that perceptions between a group of FMS patients from the USA and a group of patients from Spain, which, at first, seems interesting because these are two countries with different healthcare system models, that, in turn, could have some influence on how this disease might be perceived. However, there are some raising concern issues that must be point out, as follows: 

(lines 117-119): the procedure adopted to ensure the sample diversity should be further detailed.

(lines 121-123): the predetermined topics referred, as long as the complementary questions, could be visually represented through a figure, allowingreaders  to get a better perspective on what kind of information were target by the Authors.

(lines 188-190): since it is a qualitative cross-cultural study (line 97), what are the quantitative data obtained to run a T-test? Regarding the cross-cultural comparisons through a Chi-square Fisher's test, there isn´t any reference to this test in the results section. Actually, it could´ve been done in Table 2, where could've been analyzed if there was an association between the use of opiates and the patient's country or even a logistic regression to realize if age could be a opiates use predictor.

(lines 260-460): why not present this data like it was done in table 3, reducing sentences to keywords, becaming easier to readers realize what is the most important to retain from each one of the topics addressed by the participants.

Author Response

REVIEWER 2

The manuscript under review addresses how physical activity or exercise, its frequency, intensity, and effects are perceived by a sample of men diagnosed com FMS, and if there would exist differences on that perceptions between a group of FMS patients from the USA and a group of patients from Spain, which, at first, seems interesting because these are two countries with different healthcare system models, that, in turn, could have some influence on how this disease might be perceived. However, there are some raising concern issues that must be point out, as follows: 

Qüestion 1. (lines 117-119): the procedure adopted to ensure the sample diversity should be further detailed.

Response 1: We have added some information: 115-116 and 118-119. Also, we deleted the sentence of the reference 28 for not confusing the lectors, so we didn’t talk about ethnicity: Samples were matched as closely as possible given that perceptions and experiences of illness can be different based on factors such as age and ethnicity [28].

Montesó-Curto, P., García-Martínez, M., Gómez-Martínez, C., Ferré-Almo, S., Panisello-Chavarria, M.L., Romaguera, S., Mateu, M.L., Cubí, M.T., Colás, L.S., Usach, T.S., et al. Effectiveness of Three Types of Interventions in Patients with Fibromyalgia in a Region of Southern Catalonia. Pain Manag Nurs. 2015, 16, 642–52. DOI: 10.1016/j.pmn.2015.01.006.

Qüestion 2: (lines 121-123): the predetermined topics referred, as long as the complementary questions, could be visually represented through a figure, allowingreaders  to get a better perspective on what kind of information were target by the Authors.

Response 2: We added a Figure 1. Lines 126-143.

Qüestion 3: (lines 188-190): since it is a qualitative cross-cultural study (line 97), what are the quantitative data obtained to run a T-test? Regarding the cross-cultural comparisons through a Chi-square Fisher's test, there isn´t any reference to this test in the results section. Actually, it could´ve been done in Table 2, where could've been analyzed if there was an association between the use of opiates and the patient's country or even a logistic regression to realize if age could be a opiates use predictor.

Response 3: We deleted this sentence, we only analyzed qualitatively not quantitative.

Also we delete the word quantitative in the limitations: Hence, there are cross-cultural methodological differences, even though the quantitative and qualitative measures collected were exactly the same. Lines 453-454.

Qüestion 3: (lines 260-460): why not present this data like it was done in table 3, reducing sentences to keywords, becaming easier to readers realize what is the most important to retain from each one of the topics addressed by the participants.

Response 3: We have moved the citations to Table 3 and we have provided the categories, subcategories and the main ideas these contain so that readers can get a good understanding of the most important quotes provided by the participants. They have been added in Table 3 and on lines 276-340.

This manuscript is a resubmission of an earlier submission. The following is a list of the peer review reports and author responses from that submission.

Round 1

Reviewer 1 Report

This is an interesting study on men with fibromyalgia. The methods seem sound for a small qualitative study.

The paragraph starting at line 71 is a repeat of a paragraph above.

There are typos in the table (the word should be worse not worst

Addiction is spelled incorrectly in the table

The sentence at line 473 is not a full sentence and has no proper structure.

I think there are too many quotations. You need to synthesise these more and present representative quotations.

Author Response

Response to Reviewer 1 Comments

This is an interesting study on men with fibromyalgia. The methods seem sound for a small qualitative study

Point 1: The paragraph starting at line 71 is a repeat of a paragraph above.

Response 1: Thank you, we have removed it.

Point 2: There are typos in the table (the word should be worse not worst

Response 2: Thank you, we have changed.

Point 3: Addiction is spelled incorrectly in the table

Response 3: Thank you, we have changed.

Point 4: The sentence at line 473 is not a full sentence and has no proper structure.

“Exercising, such as walking, also offered a form of entertainment, socialization, and distraction. They reported that being distracted relieved the symptomatology, which is consistent with a previous finding [33, 34]”

Response 4: Now line 449-451.

Point 5: I think there are too many quotations. You need to synthesise these more and present representative quotations.

Response 5: Thank you we have we have eliminated some quotations leaving the most representative ones.

Reviewer 2 Report

This paper investigated the physical activity and exercise experience in Spanish and U.S. men with fibromyalgia. Interesting while further revision is needed before considering for publication.

1. In the abstract, it is mentioned "four central themes ...", while in the main text, it becomes three themes, please make sure it is consistent.

2. In the introduction part, lines 52-57 and lines 71-76 seems to be the same, please remove the redundant contents and rearrange the paragraphs. Moreover, the use of opioids seems to be a big difference in therapies for both countries, more details about the background or clinical views could be helpful to understand it.

3. The discussion on Beveridge model seems beyond the content of this paper. The Spanish participants are deeply influenced by the use of opioids, which makes them less likely to do exercises (due to drowsiness). It is a little too much to blame the system for it.

4. The citation should follow a standard pattern, for example, use Brosseau et al. (2008) in the main text.

5. There are multiple grammar mistakes, please proofread the manuscript throughout.

Author Response

Response to Reviewer 2 Comments

The revision addressed some of my concerns, and there are some minor mistakes to be corrected:

Point 1: In the abstract, line 40 doesn't make sense, please rephrase it. Moreover, the abstract should offer some insights for future study or policy implications, instead of ending with results.

Response 1: Thank you. We modify the line 40 for having sense the sentece. Line 40.

We also add a sentence for future policy impliations. Lines 43-46.

Point 2: The citation in the main text should be Brosseau et al. (2008), with year in brackets.

Response 2: Thank you, we have added in Line 68.

Point 3: lines 92-93 seems unclear to me, a 33% improvement indicates insufficient evidence? To what extent would be sufficient? Is there a standard?

Response 3: Thank you. We have complete the sentence and also we have added another sentence with a reference for the same autors (the recommendations SER). Line 79-81.

Reviewer 3 Report

(lines 35-43): I think what is exposed in these lines does not fit in with what are the precepts underlying the writing of an abstract concerning a scientific article, insofar as it does not contemplate an adequate characterization of the studied sample, a summary description of the data collection instruments, a presentation of the main results that allow answering the question that ruled the research undertaken, as well as the main conclusions this study made it possible to reach. Reading suggestion:

Andrade C. How to write a good abstract for a scientific paper or conference presentation. Indian J Psychiatry. 2011 Apr;53(2):172-5. doi: 10.4103/0019-5545.82558. PMID: 21772657; PMCID: PMC3136027.

(lines 52-53): the Authors should be cautious when stating that male patients with fibromyalgia experience delays in receiving primary care, as well as subsequent follow-up. First because the way the sentence is stated, it may convey the semantically idea of gender differences, and, second, even supported by the Paulson, Norberg & Danielson (2002) study, the Authors should have in mind study was undertaken more than 20 years and, for this reason, it may not correspond to the current reality, due to the evolution of medical-therapeutic practices, such as, for example, the eligibility criteria, illustrated by Wolfe & Rasker (2021) study, where was observed that, until 2016, the occurance of many  false positives and negatives cases, which can better explain the Authors statement in the text.

(lines 65-66): Taking into account that part of the sample under study comes from Spain, it would be interesting if the clinical position of the Spanish Society of Rheumatology were also cited and referenced, regarding the treatment of adult patients with fibromyalgia, which can be accessed through the study by Redondo et al. (2021).

Rivera Redondo, J., Díaz del Campo Fontecha, P., Alegre de Miquel, C., Almirall Bernabé, M., Casanueva Fernández, B., Castillo Ojeda, C., Collado Cruz, A., Montesó-Curto, P., Palao Tarrero, Á., Trillo Calvo, E., Vallejo Pareja, M. Á., Brito García, N., Merino Argumánez, C., & Plana Farras, M. N. (2022).
Recommendations by the Spanish Society of Rheumatology on Fibromyalgia. Part 1: Diagnosis and treatment. Reumatología Clínica (English Edition), 18(3), 131-140. https://doi.org/https://doi.org/10.1016/j.reumae.2021.02.002

(lines 71-76): why does this paragraph repeat ipsis verbis the second paragraph of the introduction? Does not make sense (...).

(lines 84-88): theses informations should be included in the abstract, once they convey what is expected to be referred on it.

Table 1. Why not identify the sample elements, in order to better characterize them, thus establishing a correspondence with the data presented in Table 2? This could be achieved using, for each of the sociodemographic variables presented in Table 1, the letters "S" (Spain) and "US" (United States), followed by the participant number indicated in Table 2, which would allow, for example, in the case of spanish participants who claimed to use opioids, analyze possible relationships between the use of this medication and their socio-demographic characteristics.

Table 2. Significant differences were observed, in the consumption of opioids, between Spanish patients (70%) and Americans (0%) with fibromyalgia, in the study sample, which arises a pertinent aspect for discussion to understand why this happens, considering that the Sociedad Española de Reumatología (SER) does not recommend the use of this range of drugs for its treatment (Redondo et al, 2022). Assuming that these are prescription drugs and therefore depend on medical recomendation, it would be important to understand the what the doctors who prescribed them think about opioids. Could it have been the patients themselves who requested their prescription? Do the doctors responsible for the prescriptions have a different opinion from their peers, regarding the treatment of fibromyalgia?

Rivera Redondo, J., Díaz del Campo Fontecha, P., Alegre de Miquel, C., Almirall Bernabé, M., Casanueva Fernández, B., Castillo Ojeda, C., Collado Cruz, A., Montesó-Curto, P., Palao Tarrero, Á., Trillo Calvo, E., Vallejo Pareja, M. Á., Brito García, N., Merino Argumánez, C., & Plana Farras, M. N. (2022). Recommendations by the Spanish Society of Rheumatology on Fibromyalgia. Part 1: Diagnosis and treatment. Reumatología Clínica (English Edition), 18(3), 131-140. https://doi.org/https://doi.org/10.1016/j.reumae.2021.02.002

(lines 473-474). the Authors should discuss why this happens, considering the Sociedad Española de Reumatología (SER) does not recommend the use of this range of drugs for the treatment of fibromyalgia (Redondo et al, 2022)? Assuming that these are prescription drugs and therefore depend on medical recommendation, it would be important to understand the reasons why 6 rheumatologists prescribed them to their patients. Could it have been the patients themselves who requested their prescription? Do the doctors responsible for the prescriptions share the Fuster & Muga´s (2018) opinion, regarding the treatment of fibromyalgia with opioids?

Rivera Redondo, J., Díaz del Campo Fontecha, P., Alegre de Miquel, C., Almirall Bernabé, M., Casanueva Fernández, B., Castillo Ojeda, C., Collado Cruz, A., Montesó-Curto, P., Palao Tarrero, Á., Trillo Calvo, E., Vallejo Pareja, M. Á., Brito García, N., Merino Argumánez, C., & Plana Farras, M. N. (2022). Recommendations by the Spanish Society of Rheumatology on Fibromyalgia. Part 1: Diagnosis and treatment. Reumatología Clínica (English Edition), 18(3), 131-140. https://doi.org/https://doi.org/10.1016/j.reumae.2021.02.002

(lines 493-494): Are there studies that support this claim? If so, they should be referenced; otherwise, some of the differences in terms of resources, culture and health policies, which allow the authors to state that a minority of the Spanish population incorporates physical exercise into their daily lives, should be listed, thus providing the reader a better perception towards the reasons why it happens.

(lines 503-504): there is a misunderstanding here, since the phrase contains a paradox, when it says that, in the United States, the Beveridge model is adopted to, right after, say that there is no public health system in this country.

(lines 523-524): some reading suggestions.

Vowles, Kevin E.a,*; McEntee, Mindy L.a; Julnes, Peter Siyahhana; Frohe, Tessaa; Ney, John P.b; van der Goes, David N.c. Rates of opioid misuse, abuse, and addiction in chronic pain: a systematic review and data synthesis. PAIN 156(4):p 569-576, April 2015. | DOI: 10.1097/01.j.pain.0000460357.01998.f1

Hauser, W., Schug, S., & Furlan, A. D. (2017). The opioid epidemic and national guidelines for opioid therapy for chronic noncancer pain: a perspective from different continents. Pain Rep, 2(3), e599. https://doi.org/10.1097/PR9.0000000000000599
pi- 523

Johnson, B., Ulberg, S., Shivale, S., Donaldson, J., Milczarski, B., & Faraone, S. V. (2014). Fibromyalgia, autism, and opioid addiction as natural and induced disorders of the endogenous opioid hormonal system.
Discov Med, 18(99), 209-220. https://www.ncbi.nlm.nih.gov/pubmed/25336035

Author Response

Response to Reviewer 3 Comments

This is an interesting study on men with fibromyalgia. The methods seem sound for a small qualitative study

Point 1: (lines 35-43): I think what is exposed in these lines does not fit in with what are the precepts underlying the writing of an abstract concerning a scientific article, insofar as it does not contemplate an adequate characterization of the studied sample, a summary description of the data collection instruments, a presentation of the main results that allow answering the question that ruled the research undertaken, as well as the main conclusions this study made it possible to reach. Reading suggestion: Andrade C. How to write a good abstract for a scientific paper or conference presentation. Indian J Psychiatry. 2011 Apr;53(2):172-5. doi: 10.4103/0019-5545.82558. PMID: 21772657; PMCID: PMC3136027.

Response 1: Thank you. We have modified the abstract, adding this information

Point 2: (lines 52-53): the Authors should be cautious when stating that male patients with fibromyalgia experience delays in receiving primary care, as well as subsequent follow-up. First because the way the sentence is stated, it may convey the semantically idea of gender differences, and, second, even supported by the Paulson, Norberg & Danielson (2002) study, the Authors should have in mind study was undertaken more than 20 years and, for this reason, it may not correspond to the current reality, due to the evolution of medical-therapeutic practices, such as, for example, the eligibility criteria, illustrated by Wolfe & Rasker (2021) study, where was observed that, until 2016, the occurance of many  false positives and negatives cases, which can better explain the Authors statement in the text.

Response 2: Thank you. We have updated the sentence to mention that it is an older finding. We are keeping the reference because Paulson et al. is one of the few studies, qualitative in this case who focus in Fibromyalgia only in men. Line 53.

Point 3: (lines 65-66): Taking into account that part of the sample under study comes from Spain, it would be interesting if the clinical position of the Spanish Society of Rheumatology were also cited and referenced, regarding the treatment of adult patients with fibromyalgia, which can be accessed through the study by Redondo et al. (2021). Rivera Redondo, J., Díaz del Campo Fontecha, P., Alegre de Miquel, C., Almirall Bernabé, M., Casanueva Fernández, B., Castillo Ojeda, C., Collado Cruz, A., Montesó-Curto, P., Palao Tarrero, Á., Trillo Calvo, E., Vallejo Pareja, M. Á., Brito García, N., Merino Argumánez, C., & Plana Farras, M. N. (2022). Recommendations by the Spanish Society of Rheumatology on Fibromyalgia. Part 1: Diagnosis and treatment. Reumatología Clínica (English Edition), 18(3), 131-140. https://doi.org/https://doi.org/10.1016/j.reumae.2021.02.002

Response 3: Thank you. We added lines 73-79.

Point 4: (lines 71-76): why does this paragraph repeat ipsis verbis the second paragraph of the introduction? Does not make sense (...).

Response 4: Thank you, we have eliminated this paragraph

Point 5: (lines 84-88): theses informations should be included in the abstract, once they convey what is expected to be referred on it.

Response 5: Thank you. We have included this information in the abstract. Lines 31-33.

Point 6: Table 1. Why not identify the sample elements, in order to better characterize them, thus establishing a correspondence with the data presented in Table 2? This could be achieved using, for each of the sociodemographic variables presented in Table 1, the letters "S" (Spain) and "US" (United States), followed by the participant number indicated in Table 2, which would allow, for example, in the case of spanish participants who claimed to use opioids, analyze possible relationships between the use of this medication and their socio-demographic characteristics.

Response 6: Thank you. We have added the Participants in Table 1. However, participants 3, 7, and 9 from Spain that are not taking opioids only have in common that they live in households of 2 to 4 persons.

Point 7: Table 2. Significant differences were observed, in the consumption of opioids, between Spanish patients (70%) and Americans (0%) with fibromyalgia, in the study sample, which arises a pertinent aspect for discussion to understand why this happens, considering that the Sociedad Española de Reumatología (SER) does not recommend the use of this range of drugs for its treatment (Redondo et al, 2022). Assuming that these are prescription drugs and therefore depend on medical recomendation, it would be important to understand the what the doctors who prescribed them think about opioids. Could it have been the patients themselves who requested their prescription? Do the doctors responsible for the prescriptions have a different opinion from their peers, regarding the treatment of fibromyalgia?.

Rivera Redondo, J., Díaz del Campo Fontecha, P., Alegre de Miquel, C., Almirall Bernabé, M., Casanueva Fernández, B., Castillo Ojeda, C., Collado Cruz, A., Montesó-Curto, P., Palao Tarrero, Á., Trillo Calvo, E., Vallejo Pareja, M. Á., Brito García, N., Merino Argumánez, C., & Plana Farras, M. N. (2022). Recommendations by the Spanish Society of Rheumatology on Fibromyalgia. Part 1: Diagnosis and treatment. Reumatología Clínica (English Edition), 18(3), 131-140. https://doi.org/https://doi.org/10.1016/j.reumae.2021.02.002

Response 7: It is an important question. The Spanish Society of Rheumatology does not recommend the use of major opioids for the treatment of pain in patients with fibromyalgia because the evidence on effectiveness is insufficient and because of the possible risk of adverse effects. But it does not say anything about minor opioids, which in this case are the ones prescribed in Spain. Lines 456-464.

Point 8: (lines 473-474). the Authors should discuss why this happens, considering the Sociedad Española de Reumatología (SER) does not recommend the use of this range of drugs for the treatment of fibromyalgia (Redondo et al, 2022)? Assuming that these are prescription drugs and therefore depend on medical recommendation, it would be important to understand the reasons why 6 rheumatologists prescribed them to their patients. Could it have been the patients themselves who requested their prescription? Do the doctors responsible for the prescriptions share the Fuster & Muga´s (2018) opinion, regarding the treatment of fibromyalgia with opioids?

Rivera Redondo, J., Díaz del Campo Fontecha, P., Alegre de Miquel, C., Almirall Bernabé, M., Casanueva Fernández, B., Castillo Ojeda, C., Collado Cruz, A., Montesó-Curto, P., Palao Tarrero, Á., Trillo Calvo, E., Vallejo Pareja, M. Á., Brito García, N., Merino Argumánez, C., & Plana Farras, M. N. (2022). Recommendations by the Spanish Society of Rheumatology on Fibromyalgia. Part 1: Diagnosis and treatment. Reumatología Clínica (English Edition), 18(3), 131-140. https://doi.org/https://doi.org/10.1016/j.reumae.2021.02.002

Response 8: No, one of the rheumatologists from Spain who has participated in our study and also his colleague see as normal to prescribe minor opiates because is what always have done following the WHO analgesic ladder where minor opioids are on the second step. They do not reach the third step. But reaching the second step is a very common clinical practice in Spain, according to what they say.

Point 9: (lines 493-494): Are there studies that support this claim? If so, they should be referenced; otherwise, some of the differences in terms of resources, culture and health policies, which allow the authors to state that a minority of the Spanish population incorporates physical exercise into their daily lives, should be listed, thus providing the reader a better perception towards the reasons why it happens.

Response 9: Beveridge system example that says that the State has to take care of your health, small auto responsibility. In Spain, most medical services are managed by the government, specifically by the Ministry of Health. This could be a possible reason why Spanish patients in this study believe that the health system should be the one to provide them with physical exercise, rather than themselves. Lines 492-506.

Point 10: (lines 503-504): there is a misunderstanding here, since the phrase contains a paradox when it says that, in the United States, the Beveridge model is adopted to, right after, say that there is no public health system in this country.

Response 10: Thanks, we have modified this accordingly.

Point 11: (lines 523-524): some reading suggestions.

-Vowles, Kevin E.a,*; McEntee, Mindy L.a; Julnes, Peter Siyahhana; Frohe, Tessaa; Ney, John P.b; van der Goes, David N.c. Rates of opioid misuse, abuse, and addiction in chronic pain: a systematic review and data synthesis. PAIN 156(4):p 569-576, April 2015. | DOI: 10.1097/01.j.pain.0000460357.01998.f1

-Hauser, W., Schug, S., & Furlan, A. D. (2017). The opioid epidemic and national guidelines for opioid therapy for chronic noncancer pain: a perspective from different continents. Pain Rep, 2(3), e599. https://doi.org/10.1097/PR9.0000000000000599

pi- 523

-Johnson, B., Ulberg, S., Shivale, S., Donaldson, J., Milczarski, B., & Faraone, S. V. (2014). Fibromyalgia, autism, and opioid addiction as natural and induced disorders of the endogenous opioid hormonal system. Discov Med, 18(99), 209-220. https://www.ncbi.nlm.nih.gov/pubmed/25336035

Response 11: Thank you, Hauser refers to major opioids, Vowles does not have a differentiation between minor o major. We have added Johnson.

Round 2

Reviewer 2 Report

The revision addressed some of my concerns, and there are some minor mistakes to be corrected:

1. In the abstract, line 40 doesn't make sense, please rephrase it. Moreover, the abstract should offer some insights for future study or policy implications, instead of ending with results.

2. The citation in the main text should be Brosseau et al. (2008), with year in brackets.

3. lines 92-93 seems unclear to me, a 33% improvement indicates insufficient evidence? To what extent would be sufficient? Is there a standard?

Author Response

(The authors gave the same response as above.)

Reviewer 3 Report

(line 139): If it is specified there was a call for students volunteers in US community, how this call process was undertaken in Spain? Once this issue is address ahead in greater detail (lines 165-167) for both countries, in my opinion, there is no need to previously reference the sample voluntarism of one of the countries, such as in the line 139.

(lines 192-194): for the quantitative data, the distribution normality was tested, in order to allow the running of a parametric test aiming to compare groups? If yes, the normality test an the p-value associated should mentioned.

(lines 535-538): considering that 5 out of 7 elements of the North American sample underwent individual interviews, while the entire Spanish sample data collection was obtained through focus groups, it might be interesting for authors to discuss their perception, regarding the interpersonal and interactive nature provided by focus groups responses as opposed to the wealth of details made possible by individual interviews, and how the implementation of these two methods, in the same research, may have contributed to the quality of the data obtained.

Suggested reading: Guest, G., Namey, E., Taylor, J., Eley, N., & McKenna, K. (2017). Comparing focus groups and individual interviews: findings from a randomized study. International Journal of Social Research Methodology, 20(6), 693-708. https://www.verityresearch.org/wp-content/uploads/2018/04/2017_Guest_ComparingFocusGroupsAndIndividualInterviews.pdf

(lines 549-553): it seems paradoxical how the long wait times and short-lived therapeutic programs can hypothetically be a reason why the Spanish patients with FMS believe that the health system should be the one to provide them with physical exercise, rather than themselves. Maybe it should the excert be rephrased to "Despite Spanish patients believe the public health system should be the one to provide them with physical exercise, rather than themselves, once medical services are accessible to the entire population, many patients, especially those seeking rehabilitation programs, experience long wait times and short-lived therapeutic programs." 

(lines 553-555): Is the hypothesis raised grounded in studies on the spanish population?

Author Response

Response to Reviewer 3 Comments

This is an interesting study on men with fibromyalgia. The methods seem sound for a small qualitative study

Point 1: (line 139): If it is specified there was a call for students volunteers in US community, how this call process was undertaken in Spain? Once this issue is address ahead in greater detail (lines 165-167) for both countries, in my opinion, there is no need to previously reference the sample voluntarism of one of the countries, such as in the line 139.

Response 1: Thank you. We have deleted line 139, and have added details for the centers where we completed the study. Of note, no call for participants was performed in Spain. Lines 137-138 and 148-149.

Point 2: (lines 192-194): for the quantitative data, the distribution normality was tested, in order to allow the running of a parametric test aiming to compare groups? If yes, the normality test an the p-value associated should mentioned.

Response 2: Thank you so much for this comment. This was an error on our part.  No quantitative data was collected or analized. We erroneously included that statement from our protocol, but later decided not to collect/analyze any quantitative data.  There is only qualitative data in our study. We deleted this sentence.

Point 3: (lines 535-538): considering that 5 out of 7 elements of the North American sample underwent individual interviews, while the entire Spanish sample data collection was obtained through focus groups, it might be interesting for authors to discuss their perception, regarding the interpersonal and interactive nature provided by focus groups responses as opposed to the wealth of details made possible by individual interviews, and how the implementation of these two methods, in the same research, may have contributed to the quality of the data obtained.

Point 4: Suggested reading: Guest, G., Namey, E., Taylor, J., Eley, N., & McKenna, K. (2017). Comparing focus groups and individual interviews: findings from a randomized study. International Journal of Social Research Methodology, 20(6), 693-708. https://www.verityresearch.org/wp- content/uploads/2018/04/2017_Guest_ComparingFocusGroupsAndIndividualInterviews.pdf 

Response 3 and 4: Thank you, we have included additional information. Lines 491-503.

Point 5: (lines 549-553): it seems paradoxical how the long wait times and short-lived therapeutic programs can hypothetically be a reason why the Spanish patients with FMS believe that the health system should be the one to provide them with physical exercise, rather than themselves. Maybe it should the excert be rephrased to "Despite Spanish patients believe the public health system should be the one to provide them with physical exercise, rather than themselves, once medical services are accessible to the entire population, many patients, especially those seeking rehabilitation programs, experience long wait times and short-lived therapeutic programs." 

Response 5: Thank you. We have included this information to clarify this idea. Lines 516-520.

Point 6: (lines 553-555): Is the hypothesis raised grounded in studies on the spanish population?

Response 6: No, this is a hypothesis that we developed.  Unfortunatley, we could not find any specific studies addressing this question, so that we have deleted it.   
